# Comparison of BODE and ADO Indices in Predicting COPD-Related Medical Costs

**DOI:** 10.3390/medicina59030577

**Published:** 2023-03-15

**Authors:** Chin-Ling Li, Hui-Chuan Chang, Ching-Wan Tseng, Yuh-Chyn Tsai, Jui-Fang Liu, Meng-Lin Tsai, Meng-Chih Lin, Shih-Feng Liu

**Affiliations:** 1Department of Respiratory Therapy, Kaohsiung Chang Gung Memorial Hospital, Kaohsiung 833, Taiwan; 2Department of Respiratory Care, Chang Gung University of Science and Technology, Chiayi 600, Taiwan; 3Chronic Diseases and Health Promotion Research Center, Chang Gung University of Science and Technology, Chiayi 600, Taiwan; 4Division of Pulmonary and Critical Care Medicine, Department of Internal Medicine, Kaohsiung Chang Gung Memorial Hospital, Kaohsiung 833, Taiwan; 5College of Medicine, Chang Gung University, Taoyuan 333, Taiwan

**Keywords:** chronic obstructive pulmonary disease, BODE index, ADO index, medical cost, prediction

## Abstract

*Background and Objectives*:The ADO (age, dyspnea, and airflow obstruction) and BODE (body mass index, airflow obstruction, dyspnea, and exercise capacity) indices are often used to evaluate the prognoses for chronic obstructive pulmonary disease(COPD); however, an index suitable for predicting medical costs has yet to be developed. *Materials and Methods*: We investigated the BODE and ADO indices to predict medical costs and compare their predictive power. A total of 396 patients with COPD were retrospectively enrolled. *Results*: For hospitalization frequencies, BODE was *R*^2^ = 0.093 (*p* < 0.001), and ADO was *R*^2^ = 0.065 (*p* < 0.001); for hospitalization days, BODE was *R*^2^ = 0.128 (*p* < 0.001), and ADO was *R*^2^ = 0.071 (*p* < 0.001); for hospitalization expenses, BODE was *R*^2^ = 0.020 (*p* = 0.047), and ADO was *R*^2^ = 0.012 (*p* = 0.179). BODE and ADO did not differ significantly in the numbers of outpatient visits (BODE, *R*^2^ = 0.012, *p* = 0.179; ADO, *R*^2^ = 0.017, *p* = 0.082); outpatient medical expenses (BODE, *R*^2^ = 0.012, *p* = 0.208; ADO, *R*^2^ = 0.008, *p* = 0.364); and total medical costs (BODE, *R*^2^ = 0.018, *p* = 0.072; ADO, *R*^2^ = 0.016, *p* = 0.098). In conclusion, BODE and ADO indices were correlated with hospitalization frequency and hospitalization days. However, the BODE index exhibits slightly better predictive accuracy than the ADO index in these items.

## 1. Introduction

Chronic obstructive pulmonary disease (COPD) is among the top three causes of death worldwide, affecting more than 300 million people worldwide.

COPD is a persistent and incurable progressive disease that is characterized by persistent lung inflammation and airflow obstruction [1,2], and the time from diagnosis to death is relatively long. Lung function decreases with age [3]. Additionally, when the severity of COPD-related dyspnea increases, COPD exacerbates rapidly to the point where hospitalization is required, and the average cost of medical care increases. In 2019, the direct medical costs for patients with COPD in the United States were USD 46.91 billion [4], and the annual economic burden of COPD disease in the European Union was EUR 48.4 billion [5]. The considerable medical costs associated with COPD have created major social, economic, and medical burdens [6,7,8].

Numerous variable tools are used to predict the exacerbation and prognoses of COPD; they include the BODE (body mass index [BMI], airflow obstruction, dyspnea, and exercise capacity), DOSE (dyspnea, obstruction, smoking status, and exacerbation), and ADO (age, dyspnea, and airflow obstruction) indices [9].

The BODE and ADO indices have been demonstrated to be accurate predictors of COPD hospitalization and mortality [10,11]. They are most commonly used as clinical tools for predicting COPD-related exacerbation, emergency department visits, hospitalizations, and deaths [12,13].

When the BODE index was divided into quartiles, its predictions for hospitalization were more accurate than those of the gold stage classification system defined in the COPD guidelines [14], and the literature indicates that BODE is associated with medical costs and that patients with higher BODE index scores had higher numbers of hospitalizations and hospital days [15,16,17].

Relative to the BODE index, the ADO index is more accurate at predicting comorbidities, mortality [18,19], and COPD exacerbation, and it indicates that COPD incidence increases rapidly with age [20]. The lung function of patients with COPD declines with age, and their number of comorbidities also increases. In particular, exacerbation- and comorbidity-related problems increase hospitalization risks and medical costs [21].

Among the components of BODE, data on exercise capacity (6-min walk distance) can be challenging to obtain, especially for older adults and people with limited mobility. The main advantage of the ADO index is that it uses widely available clinical data and is straightforward to apply.

In addition to the ability to predict COPD severity, acute exacerbation, and mortality risk, the BODE and ADO indices may also be used to predict medical costs. At present, few studies have used the BODE or ADO indices to predict COPD-related medical costs [17,22]. Numerous chronic diseases entail considerable medical costs. COPD often develops in the presence of other comorbidities, which increases its severity, leading to substantial medical costs for the entire course of the disease [23].

The main objective of the present study is to determine whether the BODE or ADO index, both of which are multidimensional assessment systems, produces more accurate medical cost predictions.

## 2. Materials and Methods

### 2.1. Study Design 

This is a retrospective study. We collected cases who had complete 6-min walk test (6MWT) records and met the GOLD guideline COPD diagnostic criteria from 31 January 2015 to 31 August 2017 in Kaohsiung Chang Gung Memorial Hospital. BODE and ADO indices were used to evaluate the COPD-related medical costs of the recipients. The study aimed to determine whether the BODE or ADO index produces more accurate medical cost predictions.

### 2.2. Study Population

Figure 1 presents a flow chart of the present study, which is a retrospective study that examined clinical outcomes. The data were extracted from the electronic database of the Kaohsiung Chang Gung Medical Center (KCGMH). We collected all patients who had received 6MWT in our institution from 31 January 2015 to 31 August 2017 (32 months; *n* = 1063). If these patients met the criteria for the diagnosis of COPD, they were enrolled in this study. The inclusion criteria and exclusion criteria that met the diagnosis of COPD are detailed below. 

The data of patients with COPD from KCGMH were used, and patients were included if they (1) had conditions that were classified under the diagnostic codes ICD-9-CM: 490–496 and ICD-10-CM: J41–J44 (*n* = 507), (2) were diagnosed with COPD on the basis of a post-bronchodilator FEV1/FVC of <70% with expiratory flow obstruction, (3) had a complete set of clinical record data that included medical cost data and the data required for using the BODE and ADO indices (*n* = 396), and (4) were classified under the BODE and ADO quartiles. The Taiwan Bureau of National Health Insurance started to use the diagnosis-related group system (Tw-DRG) in January 2010. The Tw-DRG system adopted the ICD-9-CM International Classification of Diseases before 2016 and then converted to ICD-10-CM. ICD-9-CM: 490–496 and ICD-10-CM: J41–J44 are both implemented certification disease codes for chronic obstructive pulmonary disease and related diseases. The study period is from 31 January 2015 to 31 August 2017, spanning the implementation of ICD-9-CM and ICD-10-CM in Tw-DRG.

Patients were excluded if they (1) were less than 40 years old, (2) had lung function test results that did not meet the COPD diagnostic criteria of the COPD GOLD guidelines [24], or (3) had incomplete clinical record data. On the basis of these criteria, 111 patients were excluded.

### 2.3. Clinical Variables

The BODE and ADO indices for patients with COPD were divided into four quartiles (i.e., quartile 1, 0–2; quartile 2, 3–4; quartile 3, 5–6; quartile 4, 7–10) [14,25]. The variables considered were age, gender, pack-years, pulmonary function test results, modified Medical Research Council dyspnea scale (mMRC) score, BMI, diffusing capacity for carbon monoxide, 6-min walk distance, BODE index score, ADO index score, Charlson Comorbidity Index (CCI) score, number of outpatient visits, number of hospitalizations, days of hospitalization, and medical costs.

### 2.4. Statistical Analysis 

The baseline characteristics are expressed as means and standard deviations (means ± SDs), medians (interquartile range, IQR), or N (%). The distributions for the BODE and ADO indices were evaluated through descriptive statistical analysis. The BODE and ADO quartiles were subjected to a one-way analysis of variance to compare the differences in variables on a quartile basis, and a posteriori comparisons were subsequently performed by conducting the Scheffé test. Data analysis was performed using IBM SPSS Statistics for Windows, Version 26.0. Armonk, NY, USA.

## 3. Results

We collected the BODE- and ADO-related data and clinical data of 396 patients with COPD (Table 1). The sample comprised 382 male and 14 female patients. They had an average age of 73.1 years, average smoking age of 31.7 years, average BMI (kg/m^2^) of 23.5 ± 4.1, average FEV1 (% predicted value) of 55.2 ± 18.2, average mMRC score of 1.72 ± 0.9, average 6-min walk distance of 351.9 ± 111.6 m, average BODE index score of 3.0 ± 2.1, average ADO index score of 4.9 ± 1.8, and average CCI score of 3.3 ± 2.8. On the basis of the COPD severity definitions provided in the GOLD guidelines, 58.8% of the included patients with COPD were classified as having moderate COPD, and 41.2% of them were classified as having severe or very severe COPD.

The sample was also divided into BODE and ADO quartiles. Among the BODE quartiles, quartiles 1, 2, 3, and 4 comprised 188 (47.5%), 109 (27.5%), 71 (17.9%), and 28 (7.1%) patients, respectively; among the ADO quartiles, quartiles 1, 2, 3, and 4 comprised 40 (10.1%), 124 (31.3%), 152 (38.4%), and 80 (20.2%) patients, respectively (Table 1).

Table 2 compares the medical costs for the BODE and ADO quartile groups. The BODE quartiles and ADO quartiles did not differ significantly in terms of the number of outpatient visits (BODE quartiles, *R*^2^ = 0.012, *p* = 0.179; ADO quartiles, *R*^2^ = 0.017, *p* = 0.082) or outpatient medical expenses (BODE quartiles, *R*^2^ = 0.012, *p* = 0.208; ADO quartile, *R*^2^ = 0.008, *p* = 0.364). 

Significant differences between the BODE and ADO quartiles were detected for numbers of episodes of hospitalization (BODE quartiles, *R*^2^ = 0.093, *p* < 0.001; ADO quartiles, *R*^2^ = 0.065, *p* < 0.001), days of hospitalization (BODE quartiles, *R*^2^ = 0.128, *p* < 0.001; ADO quartiles, *R*^2^ = 0.071, *p* < 0.001), and hospitalization expenses (BODE quartiles, *R*^2^ = 0.020, *p* = 0.047; ADO quartiles, *R*^2^ = 0.012, *p* = 0.179). Notably, the BODE quartiles have higher variation in the number of hospitalizations, days of hospitalization, and hospitalization expenses than the ADO quartiles. For total medical costs, the BODE quartiles (*R*^2^ = 0.018, *p* = 0.072) and ADO quartiles (*R*^2^ = 0.016, *p* = 0.098) did not differ significantly and exhibited similar levels of variation (Table 2). There is a statistically significant difference by BODE or ADO index in days of hospitalization, the number of hospitalizations, and hospitalization expenses, but BODE quartiles are more variable than ADO quartiles (Figure 2).

## 4. Discussion

In the present study, two multidimensional evaluation systems, namely the BODE and ADO indices, were applied to predict the medical costs of patients with COPD. We discovered that the BODE and ADO indices produced similar predictions for COPD-related medical costs. The BODE index was correlated to hospitalization frequencies, hospitalization days, and hospitalization expenses, but was not correlated to total medical costs, number of outpatient visits, and outpatient medical expenses. The ADO index was correlated to hospitalization frequencies and hospitalization days, but was not correlated to hospitalization expenses, total medical costs, number of outpatient visits, and outpatient medical expenses. 

For the prediction of number of hospitalizations and hospitalization expenses, the results indicate that the BODE quartiles exhibited a slightly greater level of variation relative to the ADO quartiles. The BODE quartile also exhibited a similar trend for hospitalization expenses. The results suggest that the BODE index slightly outperformed the ADO index in predicting the hospitalization expenses for patients with COPD. 

COPD is an inflammatory multisystem disease and the time from onset to death is relatively long. Most patients with COPD are older adults and patients with multiple comorbidities; age and the presence of comorbidities are key factors that increase medical costs. For disease severity, days of hospitalization under BODE quartile 3 were 10 days more than those under BODE quartile 2, and the hospitalization expenses under BODE quartile 3 were NT$ 13,000 greater than those under BODE quartile 2; days of hospitalization under ADO quartile 3 were 7.5 days more than those under ADO quartile 2, and the hospitalization expenses under ADO quartile 3 were NT$ 10,000 more than those under ADO quartile 2. Higher BODE and ADO index scores were associated with increased medical costs. Most BODE index studies have focused on predicting the exacerbation and mortality of COPD [26], and most ADO index studies have focused on exploring the effectiveness of prognosis evaluations [27]. Several studies have highlighted that the BODE and ADO index scores are associated with hospitalization rate and medical costs [16,17,22]. To date, no study has used the BODE and ADO indices to predict medical costs. Further research is required to determine whether the BODE or ADO index produces more accurate predictions with respect to medical resources.

ADO and BODE are both tools used to assess the severity of COPD and predict patient outcomes. While they may provide some insight into potential medical costs, there are many other factors that can impact COPD-related medical expenses. However, there are several studies that have examined the relationship between COPD severity and medical costs. For example, a study by Dalal et al. found that the economic burden of COPD was higher in countries with higher healthcare costs, indicating that COPD severity may impact medical expenses [28]. Another study by Ford et al. found that COPD was associated with significant medical and absenteeism costs among adults aged 18 years and older in the United States. The study estimated that the total medical costs of COPD in the US were USD 32.1 billion in 2010 and projected to increase to USD 49 billion by 2020 [29].

While ADO and BODE may not be directly related to medical costs, they can provide valuable information about COPD severity and prognosis, which may in turn impact medical expenses. For example, patients with more severe COPD may require more frequent medical visits, hospitalizations, and medication, leading to higher healthcare costs. Therefore, while ADO and BODE may be useful tools in assessing COPD severity and prognosis, a comprehensive evaluation of the patient’s medical history, current condition, and treatment plan is necessary to determine potential healthcare expenses.

Therefore, while ADO and BODE may be useful tools in assessing COPD severity and prognosis, they should not be relied upon as the sole predictor of medical costs. A comprehensive evaluation of the patient’s medical history, current condition, and treatment plan is necessary to determine potential healthcare expenses. Our study has several limitations. First, because this study is a retrospective study, prospective research is required to verify its results in a clinical setting. The prospective studies that applied the BODE and ADO indices in the context of COPD have mostly discussed the predictive power of these indices for COPD mortality and prognosis [10,19]. By contrast, few studies have examined the BODE and ADO indices in the context of healthcare burden. Second, the enrolled patients in this retrospective study were patients who met COPD criteria from the patients who had undergone the 6MWT before, rather than all patients diagnosed with COPD who then underwent the 6WMT. Most of the patients included in the study sample were male, and gender can influence outcomes, disease management, and social health costs. In Taiwan, more than 80% of patients with COPD are men [30,31]. The number of male smokers is approximately 10 times the number of female smokers, and the number of female smokers in Taiwan may be much lower than those in Western countries [32,33]. In addition, the medical record data used in the present study were obtained from a single medical center; thus, the results of the present study are not representative of all patients with COPD. Third, medical costs were divided into direct medical costs and indirect medical costs. The data we collected pertained to direct medical costs, for which a standardized payment standard is applied and the relevant hospital costs are determined by the National Health Insurance Administration [34].

## 5. Conclusions

This study compares the performance of the BODE and ADO indices in predicting COPD-related medical costs, and the results reveal that they produce similar predictions. The BODE and ADO indices are correlated to hospitalization frequencies and hospitalization days, but not correlated to total medical costs, number of outpatient visits, and outpatient medical expenses. However, the BODE index slightly outperformed the ADO index in predicting hospitalization frequencies, hospitalization days, and hospitalization costs. 

## Figures and Tables

**Figure 1 medicina-59-00577-f001:**
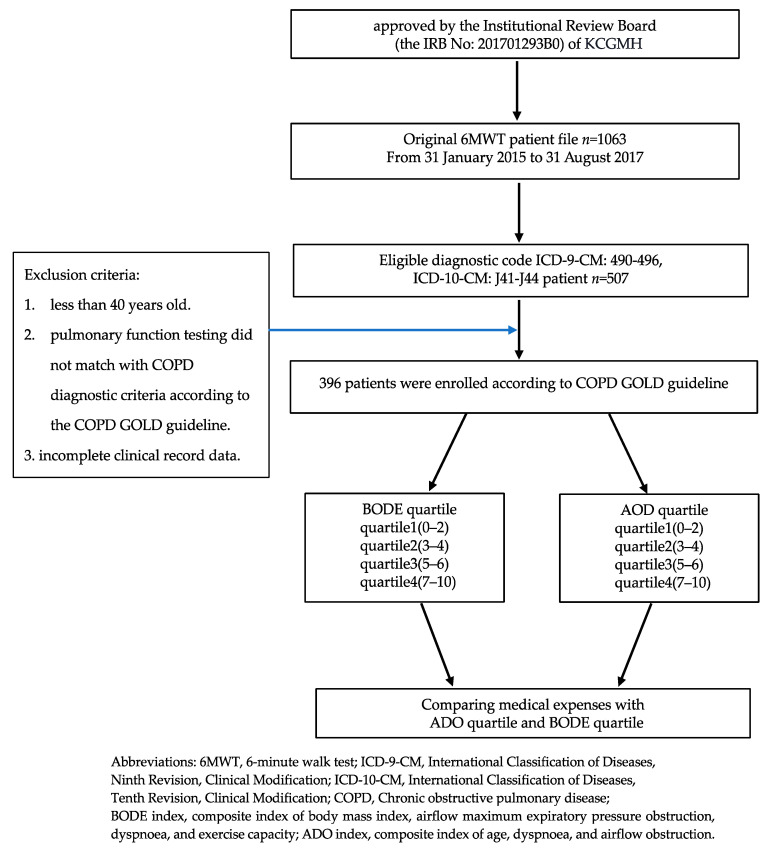
Flow chart of selected participants in this study.

**Figure 2 medicina-59-00577-f002:**
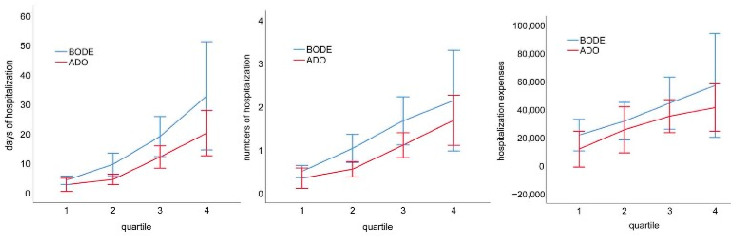
There is a statistically significant difference by BODE or ADO index in days of hospitalization, the number of hospitalizations, and hospitalization expenses, but BODE quartiles are more variable than ADO quartiles.

**Table 1 medicina-59-00577-t001:** Baseline characteristics of enrolled 396 chronic obstructive pulmonary disease (COPD) patients.

Factors	Mean ± Standard Deviation (SD) or N (%)
Male (%)	382 (96.5)
Age (years)	73.1 ± 9.5
Body-mass index (BMI)	23.5 ± 4.1
Smoking history (pack-years)	31.7 ± 18.5
FVC (% of predicted value)	79.7 ± 16.7
FEV1/FVC (%)	52.7 ± 10.6
FEV1 (% of predicted value)	55.2 ± 18.2
mMRC	1.72 ± 0.9
mMRC dyspnea scale	
Scale 0/1/2/3/4	25/133/173/56/9
6-MWD (m)	351.9 ± 111.6
GOLD stage (%)	
Mild	46 (11.6)
Moderate	187 (47.2)
Severe	140 (35.4)
Very severe	23 (5.8)
BODE INDEX	3.0 ± 2.1
ADO INDEX	4.9 ± 1.8
BODE quartile: Q1, Q2, Q3, Q4 (%) *	
quartile 1	188 (47.5)
quartile 2	109 (27.5)
quartile 3	71 (17.9)
quartile 4	28 (7.1)
ADO quartile: Q1, Q2, Q3, Q4(%) *	
quartile 1	40 (10.1)
quartile 2	124(31.3)
quartile 3	152 (38.4)
quartile 4	80 (20.2)
CCI	3.3 ± 2.8

* Quartile 1 was defined by a score of 0–2, quartile 2 by a score of 3–4, quartile 3 by a score of 5–6, and quartile 4 by a score of 7–10. Abbreviations: MRC score, Medical Research Council dyspnoea scale; FVC, forced vital capacity; FEV1, forced expiratory volume in 1 s; 6 MWD, 6-min walking distance; GOLD, Global Initiative for Chronic Obstructive Lung Disease; BODE index, composite index of body mass index, airflow maximum expiratory pressure obstruction, dyspnoea, and exercise capacity; ADO index, composite index of age, dyspnoea, and airflow obstruction; CCI, Charlson comorbidity index.

**Table 2 medicina-59-00577-t002:** BODE and ADO quartiles with COPD-related medical costs.

Classification	BODE * Quartile	Mean	Frequency or Costs NT$	*p*-Value/*R*^2^	ADO * Quartile	Mean	Frequency or Costs NT$	*p*-Value/*R*^2^
			(Mean (95% CI))				(Mean (95% CI))	
Number of outpatient visits	1	17.48	15.89–19.07	0.179/0.012	1	14.68	11.92–17.43	0.082/0.017
	2	22.92	16.09–29.74		2	17.3	15.26–19.33	
	3	20.35	17.29–23.41		3	22.68	17.64–27.73	
	4	17.04	11.49–22.59		4	19.08	16.55–21.60	
Outpatient medical expenses	1	56,263.53	45,017.65–67,509.42	0.208/0.012	1	40,843.48	30,640.11–51,046.84	0.364/0.008
	2	78,474.7	53,383.41–103,565.98		2	65,834.9	44,039.02–87,630.77	
	3	66,473.59	56,570.49–76,376.70		3	69,271.15	54,892.98–83,649.32	
	4	54,022.93	37,204.80–70,841.06		4	62,963.4	54,702.35–71,224.45	
Number of hospitalizations	1	0.51	0.37–0.65	<0.001/0.093	1	0.35	0.11–0.59	<0.001/0.065
	2	1.04	0.71–1.36		2	0.56	0.38–0.73	
	3	1.68	1.12–2.23		3	1.11	0.82–1.40	
	4	2.14	0.98–3.31		4	1.69	1.11–2.26	
Days of hospitalization	1	4.31	2.86–5.77	<0.001/0.128	1	2.75	0.53–4.97	<0.001/0.071
	2	9.69	5.86–13.52		2	4.56	2.74–6.39	
	3	19.1	12.33–25.87		3	12.16	8.37–15.94	
	4	32.89	14.55–51.24		4	20.25	12.60–27.90	
Hospitalization expenses	1	21,828.63	10608.77–33048.50	0.047/0.020	1	11,865.83	−859.49−24,591.14	0.179/0.012
	2	31,712.98	18360.53–45065.44		2	25,597.92	9280.94–41,914.90	
	3	44,593.3	26077.23–63109.36		3	35,292.59	23,548.79–47,036.39	
	4	57,114.36	20197.08–94031.63		4	41,407.2	24,540.00–58,274.4	
Total medical costs	1	78,092.16	61,640.81–94,543.52	0.072/0.018	1	52,709.3	36,714.96–68,703.64	0.098/0.016
	2	110,187.69	81,603.47–138,771.90		2	91,432.81	63,742.03–119,123.60	
	3	111,066.9	88,153.57–133,980.23		3	104,563.7	85,295.18–123,832.31	
	4	111,137.32	72,886.00–149,388.64		4	104,370.6	85,685.52–123,055.73	

* Quartile 1 was defined by a score of 0–2, quartile 2 by a score of 3–4, quartile 3 by a score of 5–6, and quartile 4 by a score of 7–10. Abbreviations: BODE index, composite index of body mass index, airflow maximum expiratory pressure, obstruction, dyspnoea, and exercise capacity; ADO index, composite index of age, dyspnoea, and airflow obstruction.

## Data Availability

The data supporting this research are available from C.-L.L.

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
