# Peer review of "Comparison of BODE and ADO Indices in Predicting COPD-Related Medical Costs"

_medicina, 2023, doi:10.3390/medicina59030577_

Round 1

Reviewer 1 Report

This is a retrospective, correlation study investigating whether the BODE and ADO indices can be used for predicting COPD-related medical expenses in Taiwan. The authors found that both indices produced similar predictions for outpatient medical expenses, however the BODE index slightly outperformed the ADO index in predicting hospitalization expenses amongst others.

The manuscript is adequately detailed with appropriate study design and description of the methods and results. I have few concerns:

1) Please add currency in Table 2 in the costs (e.g. NT$ in the column).

2) In the Discussion section, it would be beneficial to comment on the results, e.g. the highest R2-values are 0.128 for the BODE index and 0.071 for the ADO index. Would it mean that they correlate good with e.g. days of hospitalization or not? Are the BODE and ADO indices great in predicting COPD-related medical expenses? The authors have only commented on whether the BODE and ADO indices produced similar results or when BODE slightly outperformed ADO.

3) The comment above should also be reflected in the Conclusions section (both in main body and abstract).

Thank you for this manuscript!

Reviewer 2 Report

This is a retrospective study aimed at investigating BODE and ADO accuracy in predicting medical costs related to COPD.

Although the question is relevant, the conclusions of the study are based on data collected retrospectively from a single center. 

The study has several limitations, many points should be addressed and revised.

Paragraph 2.1 study design: this paragraph is not enough clear and should be reformulated.

Line 93: ICD-9- 93 CM: 490–496 and ICD-10-CM: J41–J44 should be defined.

The large majority of the sample is represented by males, with less than 5% of women included. This could represent a confounding factor. The authors explained this is a limitation, anyway the results should not be considered as general, due to this limit. 

The authors state in the paragraph 2.2 that 6 min walking tests of the included patients were analyzed, and it appears at the top of the figure 1 relative to the selection of patients. However this point has not been further considered in the manuscript.

The conclusions only partly answer the research question.

Overall, the manuscript in the present form appears not suitable for publication.

Round 2

Reviewer 2 Report

The revised version has been adequately improved.

Please correct AOD in ADO through the text.
